# Testicular histone hyperacetylation in mice by valproic acid administration affects the next generation by changes in sperm DNA methylation

**Kazuya Sakai**[1,2], **Kenshiro Hara**[1], **Kentaro Tanemura**[1] *

1 Laboratory of Animal Reproduction and Development, Graduate School of Agricultural Science, Tohoku University, Sendai, Miyagi, Japan, 2 Department of Integrative Neuroscience, National Center for Geriatrics and Gerontology, Obu, Aichi, Japan

* kentaro.tanemura.e4@tohoku.ac.jp

## Abstract

Various studies have described epigenetic inheritance through sperms. However, the detailed mechanisms remain unclear. In this study, we focused on DNA methylation in mice treated with valproic acid (VPA), an inducer of epigenomic changes, and analyzed the treatment effects on the sperm from the next generation of mice. The administration of 200 mg/kg/day VPA to mice for 4 weeks caused transient histone hyperacetylation in the testes and DNA methylation changes in the sperm, including promoter CpGs of genes related to brain function. Oocytes fertilized with VPA-treated mouse sperm showed methylation fluctuations at the morula stage. Pups that were fathered by these mice also showed behavioral changes in the light/dark transition test after maturation. Brain RNA-seq of these mice showed that the expression of genes related to neural functions were altered. Comparison of the sperm DNA methylation status of the next generation of mice with that of the parental generation revealed the disappearance of methylation changes observed in the sperm of the parental generation. These findings suggest that VPA-induced histone hyperacetylation may have brain function-related effects on the next generation through changes in sperm DNA methylation.

## Introduction

With the development of medicine and industry, the types and amounts of chemical substances in animals, including humans, have dramatically increased. Pharmaceutical and cosmetic chemicals and pesticides have enriched our lives. However, exposure to these chemicals affects the nervous, immune, and reproductive systems of animals [1]. In particular, chemicals with the potential for reproductive toxicity need to be carefully evaluated as they can significantly impact the next-generation and ecosystems. This challenge is further complicated because intergenerational effects via epigenetic changes in the reproductive system cannot be excluded [2]. Various chemicals, such as nicotine [3], arsenic [4], and vinclozolin [5],

**Data Availability Statement:** All sequencing data used in this study were deposited in the DDBJ Sequenced Read Archive (https://www.ddbj.nig.ac.jp/) under accession numbers DRA013957(https://

ddbj.nig.ac.jp/resource/sra-submission/
DRA013957), DRA013958(https://ddbj.nig.ac.jp/
resource/sra-submission/DRA013958),
DRA013959(https://ddbj.nig.ac.jp/resource/sra-
submission/DRA013959), and DRA013960(https://
ddbj.nig.ac.jp/resource/sra-submission/
DRA013960).

**Funding:** This work was supported by Grant-in-Aid
for JSPS Fellows (grant 19J21851) and Grant-in-
Aid for Scientific Research A (grant 19H01142).
The funders had no role in study design, data
collection and analysis, decision to publish, or
preparation of the manuscript.

**Competing interests:** The authors have declared
that no competing interests exist.

reportedly display intergenerational or transgenerational effects in mammals. These toxic effects of germline epigenetics are difficult to assess because they may have adverse effects on the next generation, even if they do not cause toxic effects to the exposed generation. Therefore, understanding the mechanism of the intergenerational effects of epigenetics is necessary to properly assess toxicity.

Valproic acid (VPA) is widely used in the treatment of epilepsy, bipolar disorder, and migraine headaches [6, 7]. The major pharmacological activity of VPA is the increased concentration of gamma-aminobutyric acid (GABA) in the brain due to the inhibition of GABA transaminase and succinic semialdehyde dehydrogenase activity. These inhibitions may suppress the seizures [8]. VPA also inhibits histone deacetylases (HDACs), which are key regulators of chromatin structure [9]. Although VPA is less toxic to the administered generation, its use in pregnant women is contraindicated because of its reported fetal teratogenicity and association with intellectual disability [10]. Based on these clinical findings, rodents exposed in utero to VPA have been used as animal models of autism spectrum disorder (ASD) [11–13]. Although a recent study reported that VPA triggers behavioral alterations in mice [14]. The effects of intrauterine exposure to VPA are thought to be mediated by various pharmacological effects of VPA that act directly on the fetus. However, the effects of paternal VPA ingestion are mediated by sperm and thus through germline epigenomic changes.

Epigenetics is the study of mitotically or meiotically heritable changes in gene expression or cellular phenotypes that occur without altering DNA sequences [15]. During spermatogenesis, spermatids undergo various epigenetic modifications including DNA methylation, histone modifications, and small non-coding RNA loading [16]. Although most sperm histones (approximately 95% in mice and 90% in humans, depending on the species [17]) are replaced by protamine at the end of spermatogenesis, various histone modifications are established during spermatogenesis and may be involved in the regulation of gene expression and other epigenetic modifications [18, 19]. Histone acetylation primarily alters chromatin conformation by relaxing nucleosome interactions, thereby promoting transcription [20]. Since VPA inhibits HDAC, the administration of VPA is expected to enhance the acetylation of spermatid histones, which may induce other epigenomic changes, such as DNA methylation in sperm, via structural changes in chromatin. DNA methylation is the most well-studied epigenetic modification in mammals and primarily involves the addition of a methyl group to the C5 position of cytosine. DNA methylation plays a role in regulating gene expression by recruiting proteins involved in repressing gene expression and by inhibiting the binding of transcription factors to DNA [21]. Methylated sperm DNA is demethylated immediately after fertilization. Some loci may evade demethylation and be transmitted to the next generation [22]. However, no study has examined the effects of VPA-induced histone hyperacetylation during spermatogenesis on DNA methylation in mature spermatozoa. If intergenerational effects via sperm DNA methylation is observed in VPA-treated mice, this can be used as a simple model for assessing epigenetic intergenerational toxicity or examining the mechanisms of epigenetic inheritance. Accordingly, in this study, we investigated the effects of VPA-treated sperm on the next generation of mice, focusing on DNA methylation.

## Methods

### Animals and chemicals

Male C57Bl/6N mice purchased from SLC (Shizuoka, Japan) were maintained in a temperature- and humidity-controlled room with a 12-h light/dark cycle and free access to food and water. Animals associated with tissue or cell sampling were euthanized by cervical dislocation after intraperitoneal injection of an anesthetic mixture consisting of medetomidine (0.3 mg/

kg, Meiji Seika Pharma, Tokyo, Japan), midazolam (4 mg/kg; Sandoz, Tokyo, Japan), and butorphanol (5 mg/kg, Meiji Seika Pharma); VPA was purchased from Sigma-Aldrich (St. Louis, MO, USA). All animal care and experimental procedures complied with the regulations for animal experiments and related activities of the Tohoku University. This study was approved by the Tohoku University Institutional Animal Care and Use Committee. This study was conducted in accordance with the ARRIVE guidelines.

## Treatment protocols

A total of 68 animals were divided into two groups of 34 animals each, with 4–6 animals per cage. The control group was administered a drug vehicle (saline). The VPA group was administered 200 mg/kg VPA. VPA and saline were administered daily via intraperitoneal injection to mice for 4 weeks from 6 to 10 weeks of age. The timing of this administration was determined based on the chemical effects that occurred before sexual maturity and epigenomic effects after maturity [23, 24]. Two weeks after the final dose, 38 animals were sacrificed at 12 weeks of age and used for in vitro fertilization and sperm DNA methylation analysis. This 2-week rest period facilitates the drug-affected sperm to reach the cauda epididymis and eliminates the acute effects of drug administration, enabling chronic effects to be studied. For western blot analysis, 18 animals were sacrificed 0, 3, and 7 days after the final dose. To obtain F1 offspring, 12 mice at 12 weeks of age from each group were mated with untreated, virgin, 12-week-old C57BL/6N female mice. Mating was carried out by keeping two female mice for each male mouse in the same cage. Successful mating was confirmed by the presence of sperm plugs. The resulting pups were raised to 12 weeks of age and then subjected to behavioral tests and brain RNA-seq analysis. The F1 offspring used for the behavioral experiments were individually kept from 4 weeks of age to prevent hierarchy in the home cage from affecting behavior.

## Quantification of testicular histone acetylation using multiplex fluorescent western blotting

Testes from each group were collected and homogenized in Tris-buffered saline containing a protease inhibitor cocktail (Nacalai Tesque, Kyoto, Japan). The lysates were centrifuged, and the supernatants were suspended in an equivalent volume of 2× sample buffer (Nacalai Tesque), sonicated, and boiled for 5 min. Testicular protein extracts were separated using 15% sodium dodecyl sulfate-polyacrylamide gel electrophoresis and transferred to polyvinylidene fluoride membranes. Membrane blocking was performed by incubating the samples in Blocking One (Nacalai Tesque) for 1 h at room temperature. After washing with phosphate-buffered saline, the membranes were incubated overnight at 4°C with a combination of the following antibodies: (1) rabbit polyclonal anti-histone H3 (acetyl Lys 9) antibody (diluted 1:1000, ab10812, Abcam, Cambridge, UK) and goat polyclonal anti-histone H3 antibody (diluted 1:1000, ab12079, Abcam); (2) rabbit polyclonal anti-histone H3 (acetyl Lys 27) antibody (diluted 1:1000, ab4729, Abcam) and anti-histone H3 antibody (diluted 1:1000, ab12079). The membranes were washed with phosphate-buffered saline containing 0.1% Tween 20 and treated with Alexa Fluor 555-labeled anti-rabbit and Alexa Fluor 633-labeled anti-goat secondary antibodies (diluted 1:2000) for 2 h at room temperature. After washing, multiplex fluorescent blot images were obtained using the ChemiDoc MP imaging system (Bio-Rad, Hercules, CA, USA). The fluorescence intensity of the bands was quantified using the Image Lab version 4.1 software (Bio-Rad) and normalized relative to that of histone H3.

## Collection of mouse spermatozoa

Cauda epididymides were collected from both groups, cut using micro-spring scissors, squeezed, and transferred to 1 mL of human tubal fluid (HTF) medium [25]. The medium consisted of 101.6 mM NaCl, 4.7 mM KCl, 0.37 mM $K_2PO_4$, 0.2 mM $MgSO_4 \cdot 7H_2O$, 2 mM $CaCl_2$, 25 mM $NaHCO_3$, 2.78 mM glucose, 0.33 mM sodium pyruvate, 21.4 mM, sodium lactate, 286 mg/L penicillin G, 228 mg/L streptomycin, and 5 mg/mL bovine serum albumin. After incubation for 60 min at 37.5°C in 5% $CO_2$ in humidified air, the upper layer of the medium containing motile and capacitated sperm was collected. Sperm from 10 animals in each group were used for subsequent IVF experiments, and sperm from 3 animals in each group were used for DNA extraction.

## In vitro fertilization

For oocyte collection, 4-week-old naïve C57Bl/6N females were superovulated by intraperitoneal injection of 5 IU pregnant mare serum gonadotropin (Asuka Animal Health, Tokyo, Japan), followed by 5 IU human chorionic gonadotropin (hCG; Mochida Pharmaceutical, Tokyo, Japan) 48 h later. Fifteen hours after hCG injection, the animals were sacrificed and their oviducts removed. The cumulus-oocyte complexes were collected in drops of HTF medium containing approximately $7 \times 10^5$ cells/mL sperm from 12-week-old mice from control or VPA groups. In vitro fertilization was performed by co-incubating oocytes with sperm in HTF medium drops for 4 h at 37.5°C in an atmosphere of 5% $CO_2$ in humidified air. After incubation, the oocytes were washed by gentle pipetting in potassium simplex optimized medium [26] using a glass pipette. The oocytes were transferred to new potassium simplex optimized medium drops and incubated for 72 h at 37.5°C in 5% $CO_2$ in humidified air. After incubation, morphologically normal morular embryos were selected and collected.

## Extraction of sperm DNA

Sperm DNA was extracted using a modified phenol/chloroform method [27]. Spermatozoa were suspended in lysis buffer (0.14 mM β-mercaptoethanol, 0.24 mg/ml proteinase K, 150 mM NaCl, 10 mM Tris-HCl [pH 8.0], 10 mM ethylene diamine tetraacetic acid [pH 8.0], and 0.1% sodium dodecyl sulfate) and incubated at 55°C overnight. After incubation, 400 μL of a phenol/chloroform/isoamyl alcohol (25:24:1) mixture (Nacalai Tesque) was added, and the resulting mixture was vortexed. The mixture was centrifuged at room temperature for 10 min at 12000 × g, and the upper aqueous phase was transferred to a new tube. This step of adding phenol/chloroform/isoamyl alcohol and centrifuging was repeated three times. Next, 400 μL of chloroform was added to the aqueous solution, vortexed, and centrifuged. The upper aqueous phase was transferred to a new tube containing 2 μL of ethachinmate (Nippon Gene, Tokyo, Japan), 40 μL of 3 M sodium acetate, and 800 μL of 100% ethanol and incubated for 30 min at −80°C. After incubation, the mixture was centrifuged at 4°C for 30 min at 12000 × g. The supernatant was removed, and 500 μL of 70% ethanol was added, vortexed, and centrifuged for 10 min at 12000 × g. The supernatant was removed, and the DNA pellet remaining in the tube was dried at room temperature and dissolved in Tris-ethylene diamine tetraacetic acid buffer.

## Sperm DNA methylation analysis

Sperm DNA was extracted as described above from VPA-treated mice, VPA_F1, and control mice (n = 3 each). Subsequently, DNA methylation analysis was performed using WGBS, and DNA bisulfite conversion was performed using an EZ DNA Methylation-Gold Kit (Zymo

Research, Orange, CA, USA). Further, DNA libraries were constructed using the Accel-NGS Methyl-Seq DNA Library Kit (Integrated DNA Technologies, Coralville, IA, USA) or Abclonal Scale Methyl-DNA Lib Prep Kit for Illumina (Abclonal, Tokyo, Japan) according to the manufacturer's instructions. The DNA libraries were used for 150 bp paired-end sequencing on a Novaseq 6000 platform (Illumina, San Diego, CA, USA). Adapter trimming was performed using the Trim Galore 0.5.0 program (RRID: SCR_011847) and Trimmomatic program [28]. The resulting reads were quality checked using FastQC version 0.11.7 [29]. The reads were aligned to the mouse genome mm10 using Bismark version 0.22.3 [30] with default parameters. Gene annotation and methylation level calculations were performed using the methylKit package [31] in R (https://www.r-project.org/). Promoters were defined as regions 2 kb upstream of the transcription start sites. The DMCs were identified as cytosines in the promoter regions whose methylation rate showed a P-value $< 0.05$, as determined by t-test, and the mean methylation rate difference was $\geq 20\%$. Similarly, regions that contained at least 10 CpGs within 300 bp and whose average difference in methylation rates was $\geq 20\%$ were extracted as DMRs. The genomic regions rich in acetylated histones K9 and K27 located in the vicinity of the DMRs ($\pm 1$ kb) were identified using the Peak Browser of ChIP-Atlas [32], where the threshold for significance was set to 50 (q-value $< 1E-05$) and cell type as mouse male germ line (testis, male germ cells, spermatogonia, spermatogenic cells, round spermatids, and spermatids). Pathway analysis using IPA software version 68752261 (QIAGEN, Valencia, CA, USA) was performed for genes containing DMC in the promoter region.

## Morula DNA methylation analysis

DNA methylation analysis was performed by the WGBS method using morulas derived from VPA-treated and control mouse sperms. In this experiment, 250–300 morulas derived from three to four mouse sperms per sample were used. Morula DNA was extracted using NucleoSpin Tissue XS (TaKaRa Bio, Shiga, Japan). Next, DNA bisulfite conversion, library construction, and sequencing were performed using the same kits and methods as in the sperm DNA methylation analysis experiments described above. The resulting reads were adapter-trimmed, quality checked, and aligned with the mouse reference genome mm10. Methylation rates were then calculated for each CpG, for which data existed in all samples, using the methylKit package in R. We compared these CpG methylation rates and examined the similarities between samples using principal component analysis (PCA).

## Behavioral test

Fifteen male F1 mice were selected from each group to have the smallest variance in body weight at 12 weeks of age and were subjected to behavioral tests. Behavioral tests included the open field, light/dark transition, and context/cued fear conditioning tests, as previously described [33, 34]. The procedures and equipment used in the behavioral experiments described below are minor modifications to those used in a previous report by Saito et al [33]. The measured values and images were analyzed using Image OF2, Image LD2, and Image FZ2 software (O'Hara & Co., Ltd.) developed using the public domain ImageJ program [35]. The experimental tests were conducted between 10:00 and 15:00. The experiments were performed in a soundproof box (78 × 63 × 65 [H] cm) made of white-colored wood, which was equipped with an audio speaker and a light source. Background noise of approximately 50 dB was applied during the experiments. After each trial, the apparatus was cleaned with water and wiped dry.

**Open field test.**   Locomotor activity was measured for 10 min using an OF apparatus made of white plastic (50 × 50 × 30 [H] cm). The LED light system was positioned approximately 50

cm above the center of the field (25 lx at the center of the field). The behavior was measured using a charge-coupled device (CCD) camera positioned above the center of the apparatus.

**Light/Dark transition test.** The apparatus consisted of a cage ($21 \times 42 \times 25$ [H] cm) divided into two chambers by a partition with an opening. One chamber was made of white plastic and was brightly illuminated (250 lx, light box). The other chamber, made of black plastic, was dark (5 lx, dark box). Behavior was measured using a CCD camera positioned above each chamber. The mice were placed in a dark box and allowed to move freely between the two chambers through the opening for 5 min.

**Contextual/Cued fear conditioning test.** The apparatus consisted of a conditioning chamber (test chamber; $17 \times 10 \times 10$ [H] cm) made of clear plastic with a ceiling. The chamber floor had stainless steel rods (2 mm diameter) spaced 5 mm apart that delivered an electric foot to the feet of the mice. The inner wall of the chamber was covered with black and white plastic strips. The LED light system was positioned approximately 50 cm above the chamber (50 lx at the center of the floor). Behavior was measured using a CCD camera positioned above the center of the chamber. During the conditioning trial, mice were individually placed in the conditioning chamber and, after 40 s, were given six tone-shock pairings (20 s of tone at 65 dB, directly followed by 2 s of 0.08 mA electric shock), each separated by 60 s. The mice were then returned to their home cage. The next day, as a contextual fear test, the participants were returned to the conditioning chamber for 6 min without tone or shock. The day after that, for the cued fear test, they were placed in a novel chamber (with a different design and lacking plastic black and white stripes and stainless steel rods). After 3 min, a conditioning tone (with no shock) was presented for 3 min. The freezing response of the mice was measured using Image FZ2 as a consecutive 2-s period of immobility. Freezing rate (%) was calculated as follows: (freezing/session time) $\times$ 100.

## Brain RNA-seq

RNA-seq was performed to obtain the brain RNA expression profiles of the F1 mice subjected to behavioral experiments. Four mice in each group were sacrificed after the behavioral experiments. Total RNA was extracted from the brain tissue (whole brain, excluding the olfactory bulb and cerebellum) using the NucleoSpin RNA kit (TaKaRa Bio). The concentrations of the RNA samples were determined using a NanoDrop ND-1000 spectrophotometer (Thermo Fisher Scientific, Waltham, MA, USA). The quality was checked using the 5200 Fragment Analyzer System (Agilent, Santa Clara, CA, USA) and Agilent HS RNA kit (Agilent). Sequence libraries were then constructed using the MGIEasy RNA Directional Library Prep Set (MGI Tech; Shenzhen, China). The quality of the library was checked using the 5200 Fragment Analyzer System with dsDNA 915 Reagent Kit (Agilent). One hundred base-pair paired-end sequencing was performed on the DNBSEQ-G400 platform using the DNBSEQ-G400RS high-throughput sequencing kit (MGI Tech). The resulting raw reads were quality-checked using the Sickle program. Adapter trimming was performed using the Cutadapt 1.16 program. Trimmed reads were mapped against mm10 using the HISAT2 program, followed by annotation and transcript quantification using StringTie. Transcript expression levels were compared to detect differentially expressed genes (DEGs) using edgeR; DEGs were defined as genes with P-values $< 0.05$ and two-fold or greater expression variation by exactTest. These were used for pathway analysis using IPA software version 68752261 (QIAGEN).

## Statistical analyses

Data from western blotting and behavioral analysis are presented as the mean ± S.E. or the mean + S.E., using Student's t-test for comparisons. Some behavioral analysis data are depicted

using box-and-whisker plots (consisting of the median, interquartile range, minimum and maximum values, and outliers). For sequencing data, we used a t-test as well as the exactTest in the edgeR package of R. Statistical significance was set at P < 0.05. All statistical analyses, except pathway analysis and graphic drawing, were performed in R. P-values in the pathway analysis were calculated using the IPA software.

## Results

### Effect of VPA administration on testicular histone acetylation

During the period of VPA or saline administration, all animals were healthy and had no clinical abnormalities, including testicular toxicity. To determine whether VPA administration to mice increases testicular histone acetylation, a comparative analysis of acetylated histones H3K9 and H3K27 was performed using multiplex fluorescent western blotting (Fig 1A), and the unprocessed blots are depicted in S1 and S2 Raw images. Acetylation of both H3K9 and H3K27 was significantly increased compared to that in the controls in the testes collected immediately after VPA administration. However, acetylation was reduced to the same level as that in the controls in testes collected 3 and 7 days after the end of treatment (Fig 1B). These findings indicate that VPA administration transiently increases histone acetylation in the mouse testes.

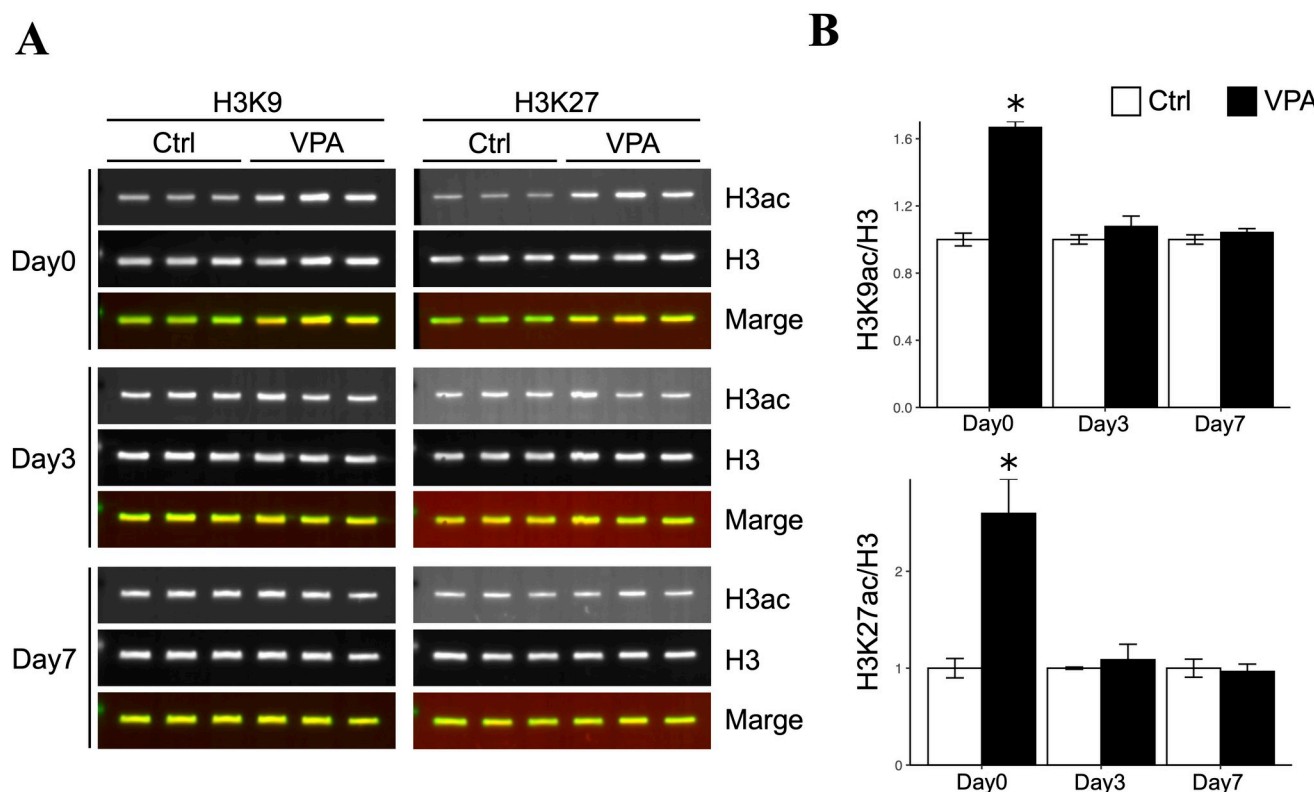

**Fig 1. Testicular histone acetylation levels determined using multiplex fluorescent western blotting.** (A) All blotting images of the control and VPA-treated groups are depicted. On the left is histone H3K9 and on the right the bands of histone H3K27, and from the top row, immediately after treatment with VPA or saline, 3 days later, 7 days later, the histone acetylation bands, histone H3 bands, merged bands images. Histone acetylation and histone H3 bands images are shown in monochrome. (B) The bars show the fluorescent signals of histone acetylation at each sampling point. The signals are standardized to the histone H3 signal, with value relative to the control group as 1. Data are shown as mean ± S.E. (n = 3, *P < 0.05).

## Sperm DNA methylation analysis of VPA-treated mice

To investigate the effects of VPA administration on DNA methylation in mouse spermatozoa, a genome-wide DNA methylation analysis was performed using whole-genome bisulfite sequencing (WGBS). Significant methylation changes were detected in the cytosines of 1863 CpG contexts in the promoter region. Of these, 846 were hypermethylated, and 1017 were hypomethylated in the VPA-treated group (Fig 2A). Pathway analysis of the downstream genes of these promoters was then performed. This analysis revealed that many genes were associated with brain function-related pathways that included "axonal guidance" and "amyloid processing" in the downstream genes of promoters, containing hypermethylated CpGs. Various pathways that included "sulfate activation for sulfonation" and "RAR activation" were detected in the downstream genes of promoters, including hypomethylated CpGs (Fig 2B). Similarly, we searched for upstream regulators of these genes using Ingenuity pathways

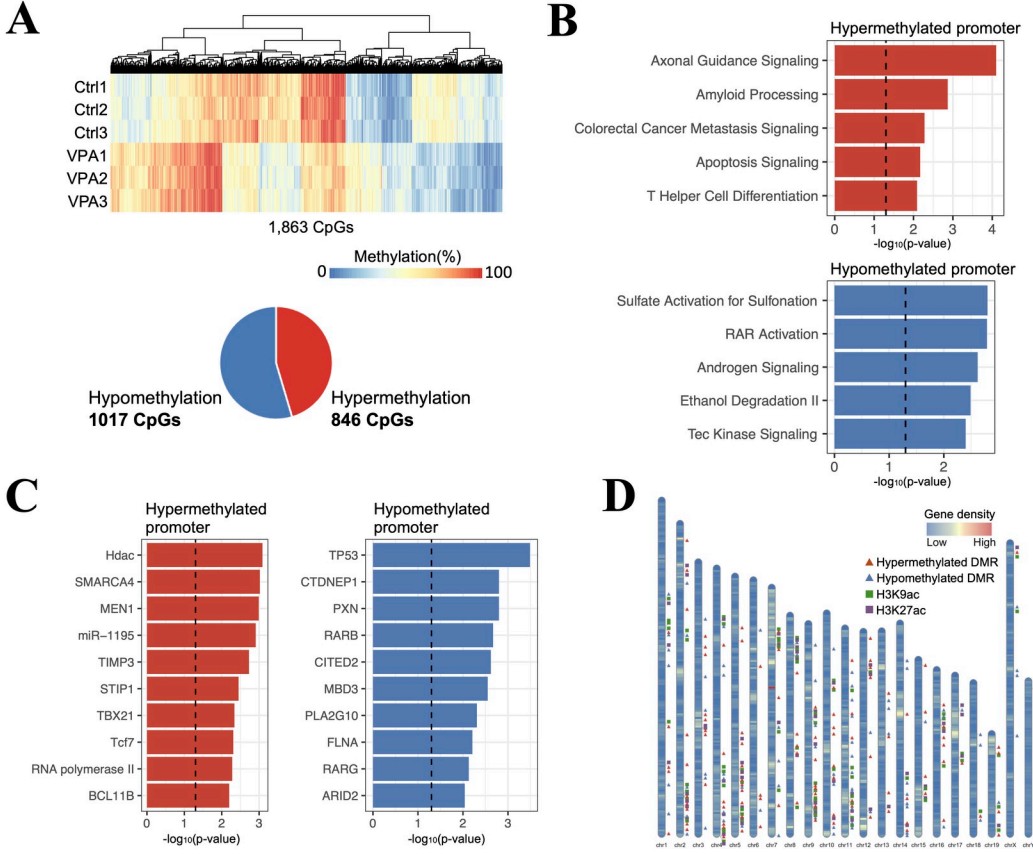

**Fig 2. Sperm DNA methylation analysis of VPA-treated mice using whole-genome bisulfite sequencing (WGBS).** (A) Heat map showing cytosines in the CpG context that were significantly changed in promoter methylation. The number of significantly hypermethylated and hypomethylated cytosines in the VPA-treated group compared with the control group is shown in the pie chart. P-value < 0.05 by t-test and whose mean methylation rate difference was ≥ 20% were considered significant (n = 3). (B) Top canonical pathways generated by Ingenuity Pathway Analysis (IPA), using the downstream genes of promoters with altered methylation as input. The dashed line indicates $-\log_{10}$ (0.05). (C) Upstream regulators predicted by IPA, using downstream genes of the promoters with altered methylation as input. (D) Positions of differentially methylated regions (DMRs) found in the VPA-treated group compared with those of the control group are indicated by triangle markers on the ideogram. The positions of acetylated histone H3 (K9 and K27) which have been reported to be present in the vicinity of DMRs (±1 kb) in the male germline are also indicated by rectangular markers on the ideogram. The regions that contained at least 10 CpGs within 300 bp and whose average difference in methylation rate was ≥ 20% were extracted as DMRs. Genomic regions with a score of 50 or higher in the ChIP-Atlas Peak Browser were designated as acetylated histone regions.

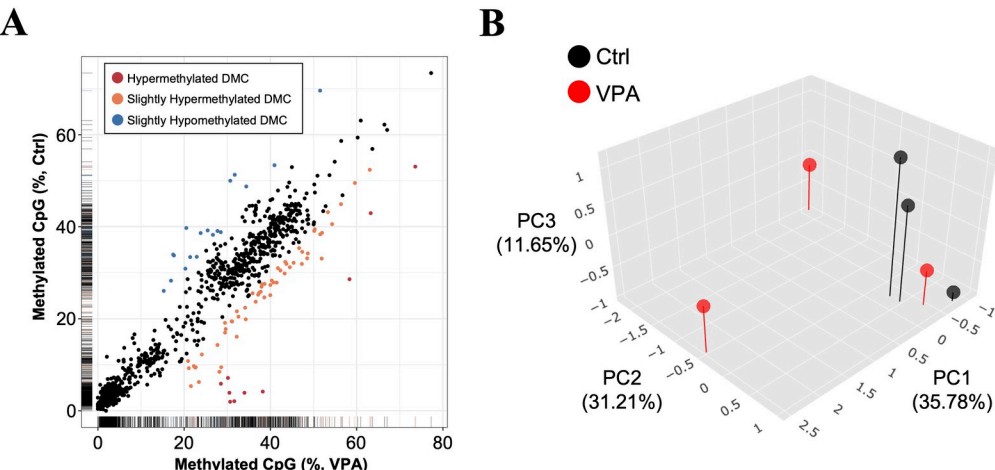

**Fig 3. Morula DNA methylation analysis using whole-genome bisulfite sequencing.** (A) Comparison of CpG methylation levels in morula derived from sperm of control and VPA-treated groups. Only CpGs with sequencing reads common to all samples are shown (1109 CpGs in total). The average methylation levels are indicated. CpGs with > 20% hypermethylation compared with that of the control group are shown in red, those with > 10% hypermethylation in orange, and those with > 10% hypomethylation in blue. (B) Principal component analysis (PCA) using methylation reads distribution for each sample.

analysis (IPA) and found that many genes with hypermethylated CpGs in their promoters were regulated by HDAC. Tumor protein P53 (TP53) showed the highest P-value as an upstream regulator in genes with promoters containing hypomethylated CpG sites (Fig 2C). We explored regional methylation changes and found 187 differentially methylated regions (DMRs) across all chromosomes. We then searched for acetylated histone H3(K9 and K27)-rich genomic regions in the vicinity of these DMRs (±1 kb) using public ChIP-seq data provided by ChIP-Atlas and found that many DMRs were in the vicinity of acetylated histone H3 (Fig 2D). Collectively, these results indicate that sperm DNA methylation is altered by VPA administration. In particular, the results suggest that hypermethylation in the promoter regions can be attributed to the inhibition of HDACs by VPA and that many methylation changes occur in genes related to brain function.

## DNA methylation analysis of morula derived from VPA-treated mice sperm

To determine whether sperm-derived DNA methylation changes were maintained in the embryo after fertilization, we performed methylation analysis of the morula produced by conventional in vitro fertilization (IVF). The degree of methylation of each CpG was similar between the groups, with low to moderate methylation rates (mean methylation in the control group: 19%, VPA-treated group: 20%). However, a relatively large number of hypermethylated CpGs was also found in embryos derived from VPA-treated mouse sperm (Fig 3A). Principal component analysis (PCA) was then performed on all the samples to examine the uniformity of the methylation data across samples. The VPA group data tended to be scattered, whereas the control group had similar data (Fig 3B).

## Behavioral analysis of F1 mice

To examine the effects of VPA-treated mouse sperm on the next generation, behavioral tests were performed on each group of F1 male mice. Forty-five F1 male mice were obtained from

the control group and 52 from the VPA-treated group, all of which were healthy. Fifteen animals from each group were selected for the behavioral experiment to minimize the variance in body weight. The next generation of control mice (Ctrl_F1) weighed an average of 24.7 g, and the next generation of VPA-treated mice (VPA_F1) weighed an average of 25.0 g. The behavioral tests included an open field, light/dark transition, and fear conditioning tests. In the open field test, no significant differences were observed between VPA_F1 and Ctrl_F1 mice (Fig 4A). In the light/dark transition test, the time spent in the light box was significantly increased in VPA_F1 compared with that in Ctrl_F1 (Fig 4B). In the fear conditioning test, no significant differences were observed between the groups in any of the experiments (conditioning, contextual, and cued tests) performed over a 3-day period (Fig 4C).

## Brain gene expression analysis

The slight changes in the behavioral tests prompted brain RNA-seq analysis in both F1 groups. A variety of gene expression changes were evident in these brains. In the VPA_F1 group, 20 genes were found to be significantly (FDR < 0.05) upregulated, and 4 genes were significantly downregulated (Fig 5A). Pathway analysis using these genes detected a majority of the top pathways related to neuronal function, including endocannabinoid neuronal synapse pathway, dopamine-DARPP32 feedback in cAMP signaling, and CREB signaling in neurons (Fig 5B).

## Sperm DNA methylation analysis of F1 mice

To investigate whether the changes observed in the F1 generation are transmitted to the F2 generation, we performed DNA methylation analysis of sperm obtained from the F1 mice and compared the results with those of the F0 generation. The CpGs that showed significant methylation changes in the promoter region of the F0 generation samples showed no significant changes in that of the F1 generation samples, as the differences were resolved in both Ctrl_F1 and VPA_F1 groups (Fig 6A). Similarly, only 13 DMR regions were detected in the F1 generation, compared with nearly 200 regions in the F0 generation (Fig 6B).

## Discussion

Some studies have suggested that nicotine [3], arsenic [4], and other substances affect the next generation through DNA methylation. However, the underlying mechanism remains unclear. We attempted to induce relaxation of chromatin structures through testicular histone hyperacetylation by administering VPA, an HDAC inhibitor. We hypothesized that the relaxation of chromatin structure during spermatogenesis would disturb DNA methylation changes in the sperm and that some of these changes would be inherited by the next generation.

As hypothesized, VPA administration induced histone hyperacetylation in testes. This hyperacetylation was comparable with that of the control group at least 3 days after the end of treatment, indicating that the effect of VPA was transient. Since the treatment in this study was daily administration of VPA for 4 weeks, it is expected that spermatozoa formed during this period differentiated while maintaining histone hyperacetylation. Considering that the transit time of the epididymis in mice is approximately 10 days [36] and that the time required for spermatogenesis is 34.5 days [37], the spermatozoa used for DNA methylation analysis in this study are presumed to have included highly acetylated histones during most of the formation process. The DNA methylation analysis revealed transient histone hyperacetylation during spermatogenesis, and DNA methylation changes remained in the sperm. Many of the DMRs we found are located near acetylated histone regions, suggesting that HDAC inhibition by VPA is linked to DNA methylation changes. The number and distribution of promoter differentially methylated cytosines (DMCs) and genome-wide DMRs suggested that disturbed,

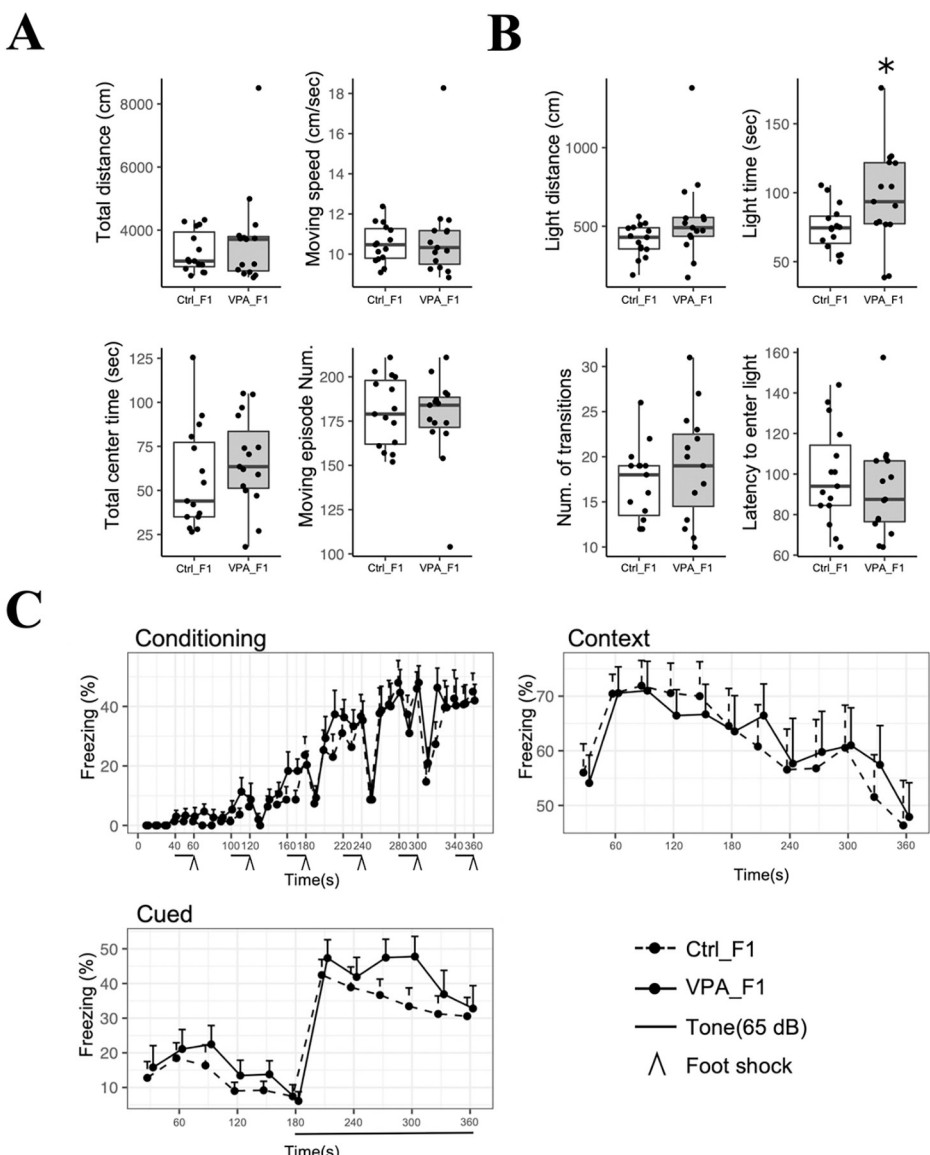

**Fig 4. Results of behavioral tests for VPA-treated F1 and control group F1 mice.** (A) Results of the open field test are shown in the box-and-whisker plots with overlaid dot plots representing individual data (*P < 0.05, Student's t-test, n = 15 per group). The median is marked by a bold line inside the box. The boxes show the 25th–75th percentiles. The whiskers outside the box indicate the range from minimum to maximum values. Data outside the box + 1.5 × interquartile range (IQR) are shown as outliers. Distance traveled (cm), time spent in the center area (sec), average speed (cm/s), and total moving episode number are shown. (B) Results of the light/dark transition test. Distance traveled in the light-box (cm), time spent in the light box (sec), number of times the two boxes have been moved back and forth, time to first enter the light box after the start of the test (sec) are shown. (C) Results of the contextual/cued fear conditioning test expressed as time courses of the freezing scores (%, mean + S.E., n = 15 per group). For the conditioning trial and contextual fear test, the total freezing time was used for statistical analysis by Student's t-test. For the cued fear test, the total freezing time in the second half (180–360 s) was used.

rather than directional, methylation changes occurred in the VPA-treated sperm genome. Notably, our pathway analysis suggested that brain function-related genes were enriched downstream of the promoter where hypermethylated DMCs were located. Moreover, in the upstream regulator analysis, HDAC showed high P-values, suggesting that CpG hypermethylation may contain directional changes attributable to VPA administration.

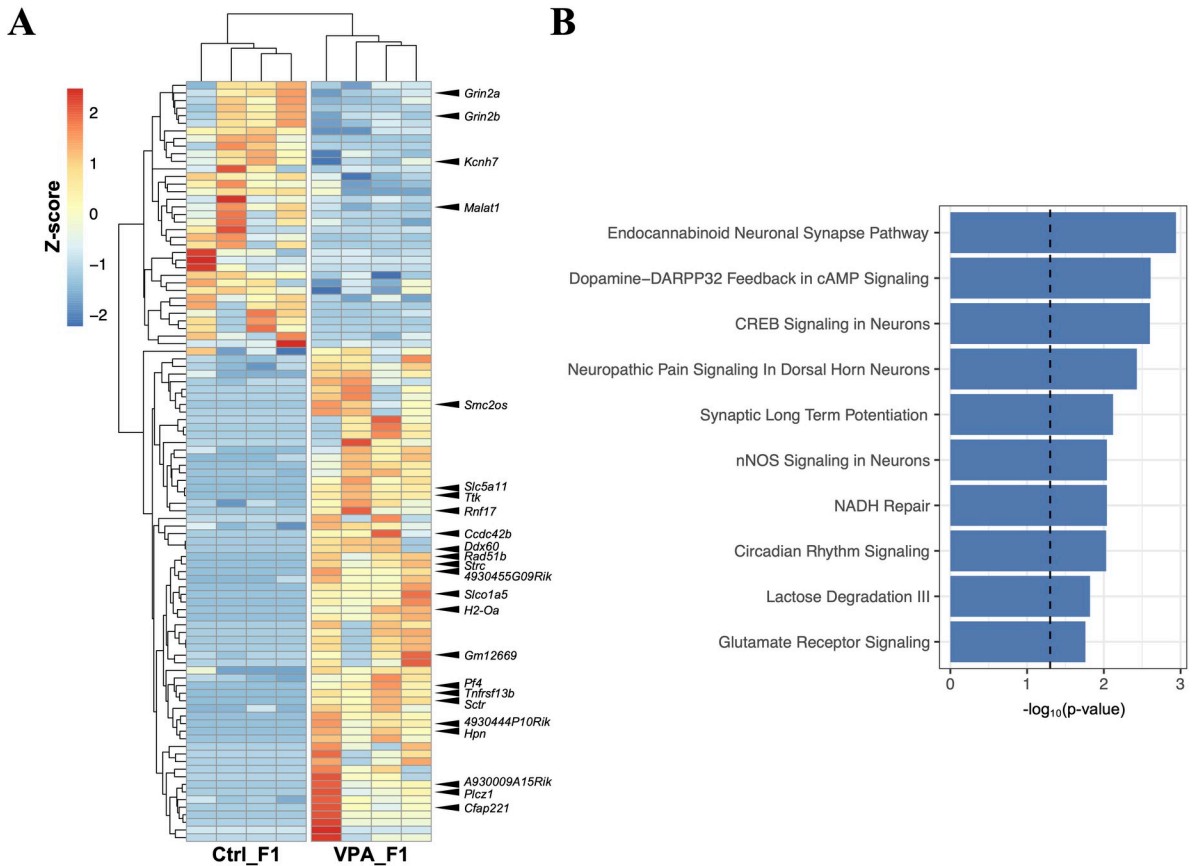

**Fig 5. Brain RNA-seq for VPA-treated F1 (VPA_F1) and control group F1 (Ctrl_F1) mice.** (A) Genes with P-value < 0.05 and two-fold change are shown in the heatmap (n = 4). q-value ≤ 0.05 is indicated by arrowheads, and the gene name is attached. (B) Top canonical pathways generated by Ingenuity Pathway Analysis (IPA), using the genes with P-values < 0.05 and two-fold or greater expression as input. The dashed line indicates $-\log_{10}(0.05)$.

The sperm-derived genome undergoes active demethylation in the male pronucleus immediately after fertilization via Tet3-mediated DNA hydroxylation [38]. However, some regions may escape demethylation and be transmitted to the next generation [22]. We performed IVF using VPA-administered mouse sperm, cultured them to the morula stage, and analyzed DNA methylation. Since the maternally derived genome has diluted DNA methylation upon cell division due to the weak function of the maintenance DNA methyltransferase 1 (DNMT1), morula DNA is expected to exhibit poor methylation [39]. RNA-seq analysis did not provide a satisfactory amount of data owing to low inputs, and identification of statistically evident differentially methylated loci was not possible. However, as hypothesized, an overall low methylation rate was evident. In addition, some CpGs from the VPA-administered mouse-derived morula remained slightly (but still highly) methylated, and the methylation rates tended to vary among samples. Although these differences are not necessarily keys to epigenetic inheritance, VPA treatment likely causes qualitative differences in sperm.

We observed behavioral changes in VPA_F1. The increase in light duration in the light/dark test observed in VPA_F1 mice suggests a decrease in anxiety-related behavior. Our RNA-seq analysis identified many differentially expressed genes (DEGs) in the brain of VPA_F1 mice, and our pathway analysis detected many neural function-related pathways. Genes involved in many of these pathways included *Grin2a* and *Grin2b*, which encode N-methyl-D-

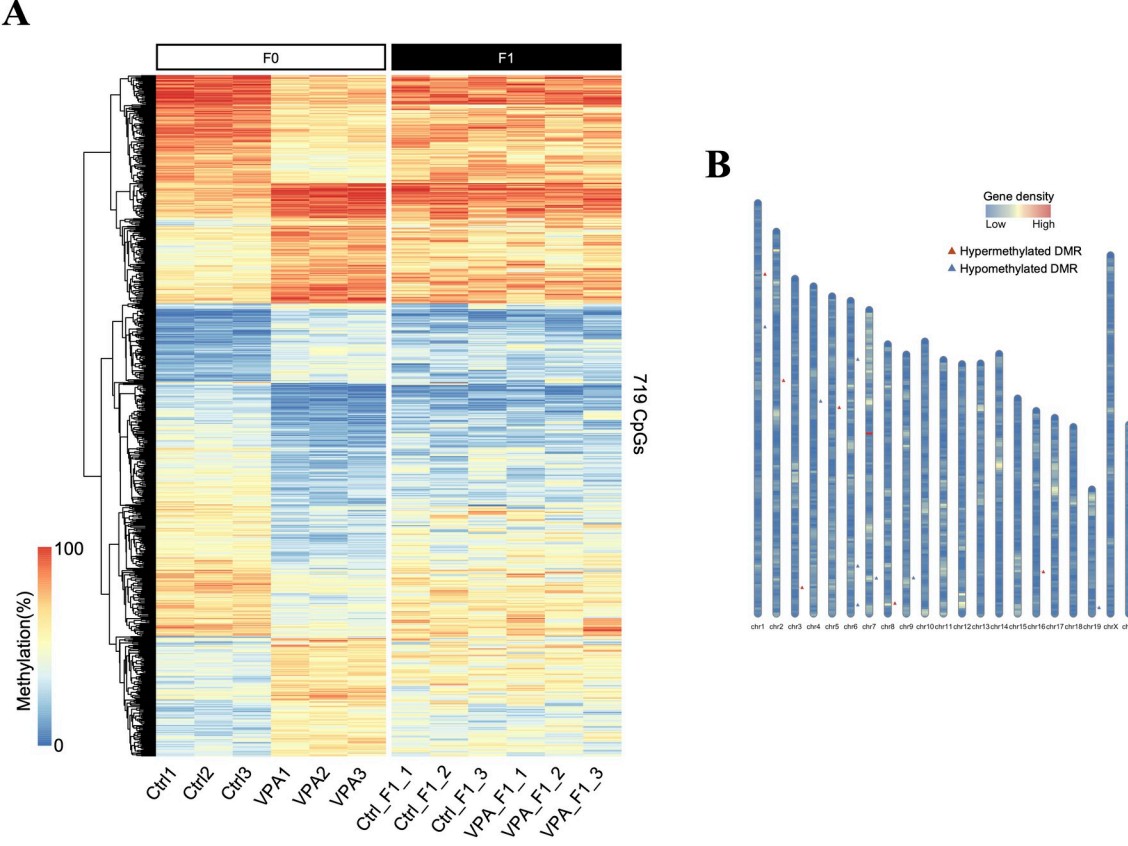

**Fig 6. Sperm DNA methylation analysis of VPA-treated F1 mice using whole-genome bisulfite sequencing.** (A) Heat map comparison of all samples of sperm DNA methylation levels, including data from the next generation of VPA-treated mice (VPA_F1) and next generation of control group mice (Ctrl_F1) at promoter CpG loci where significant changes were observed in the VPA-treated generation (F0 generation, shown in Fig 2A). The left six columns indicate the F0 generation and right six columns the F1 generation. (B) Positions of differentially methylated regions found in the VPA_F1 group compared with those of the Ctrl_F1 group are indicated by markers on the ideogram.

aspartate (NMDA) receptor subunits, both of which were significantly downregulated in the VPA_F1 group. Further, NMDA receptors play an important role in various neural activities such as synaptic transmission, synaptic plasticity, and neurodevelopment in the central nervous system [40]. Since the inhibition or mutation of NMDA receptors induces various behavioral abnormalities [41], we suggest that the behavioral changes observed in this study also involve a reduction in *Grin2a* and *Grin2b* expression. However, whether these changes affect protein expression or these gene expression changes are due to DNA methylation remains unresolved and should be investigated.

Several studies have reported the epigenomic effects of environmental factors over multiple generations [42, 43]. However, in our study, methylation changes in the sperm DNA of VPA-treated mice mostly disappeared in the F1 generation. Thus, if the intergenerational effects observed in our study were due to DNA methylation, transmission across multiple generations would be unlikely. We hypothesized that VPA treatment relaxes chromatin and increases susceptibility to epigenomic changes, but does not cause directional changes in DNA methylation. Therefore, DNA methylation changes that would persist for multiple generations would not occur.

In this study, we demonstrated that VPA administration to mice alters sperm DNA methylation through histone hyperacetylation during spermatogenesis, which affects the next generation. We believe that this phenomenon is due to the synergistic effects of chromatin relaxation, which promotes epigenetic changes, and the pharmacological effects of VPA itself, such as HDAC inhibition and involvement in GABA signaling. Since a recent study suggests that VPA affects DNA modification independently of histone acetylation, future studies should validate our results with HDAC inhibitors other than VPA, such as trichostatin A [44]. If the changes observed in this study are due to enhanced epigenomic susceptibility caused by chromatin relaxation, inducing stronger epigenomic inheritance is possible by combining the treatment with environmental stimuli, which are known to cause epigenomic inheritance. Future studies must focus on a more detailed analysis of DNA methylation and gene expression during embryonic and brain development and aim to confirm that the changes in brain gene expression observed in F1 individuals contribute to behavioral changes as effects on functional proteins.

## Supporting information

**S1 Raw image. Unprocessed blots for acetylated histones H3K9.** The three lanes from the left (lane number 1–3) are the bands of the control group and the three lanes from the right (lane number 4–6) are the bands of the VPA-treated group.
(PDF)

**S2 Raw image. Unprocessed blots for acetylated histones H3K27.** The three lanes from the left (lane number 1–3) are the bands of the control group and the three lanes from the right (lane number 4–6) are the bands of the VPA-treated group.
(PDF)

## Acknowledgments

We thank Dr. Saito for allowing us to cite some of the descriptions from his paper in the section on methods of behavioral experiments.

## Author Contributions

**Conceptualization:** Kazuya Sakai, Kentaro Tanemura.

**Data curation:** Kazuya Sakai.

**Formal analysis:** Kazuya Sakai.

**Funding acquisition:** Kentaro Tanemura.

**Investigation:** Kazuya Sakai.

**Methodology:** Kazuya Sakai.

**Supervision:** Kenshiro Hara, Kentaro Tanemura.

**Validation:** Kazuya Sakai.

**Visualization:** Kazuya Sakai.

**Writing – original draft:** Kazuya Sakai.

**Writing – review & editing:** Kazuya Sakai.

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
