## [Decision Letter · Decision Letter 0]

22 Nov 2022

PONE-D-22-26282Testicular histone hyperacetylation in mice by valproic acid administration affects the next generation by changes in sperm DNA methylationPLOS ONE

Dear Dr. Tanemura,

Thank you for submitting your manuscript to PLOS ONE. After careful consideration, we feel that it has merit but does not fully meet PLOS ONE’s publication criteria as it currently stands. Therefore, we invite you to submit a revised version of the manuscript that addresses the points raised during the review process. While the results presented are very interesting, there are some serious questions that need to be addressed. The major being the link between DNA methylation and histone hypermethylation. There is no direct evidence to establish this fact. Acetylation of tubulin and H4 are know to play crucial roles in spermatogenesis. A direct relation between the acetylation of these two with the observed effects has to be provided. Further, the relevance of behavioural studies needs to be justified further.

We look forward to receiving your revised manuscript.

Kind regards,

Suresh Yenugu

Academic Editor

PLOS ONE

Journal Requirements:

“This work was supported by Grant-in-Aid for JSPS Fellows (grant 19J21851) and Grant-in-Aid for Scientific Research A (grant 19H01142).”

Reviewers' comments:

Reviewer's Responses to Questions

**Comments to the Author**

1. Is the manuscript technically sound, and do the data support the conclusions?

Reviewer #1: No

Reviewer #2: Yes

Reviewer #3: Yes

2. Has the statistical analysis been performed appropriately and rigorously? 

Reviewer #1: Yes

Reviewer #2: Yes

Reviewer #3: I Don't Know

3. Have the authors made all data underlying the findings in their manuscript fully available?

Reviewer #1: Yes

Reviewer #2: Yes

Reviewer #3: Yes

4. Is the manuscript presented in an intelligible fashion and written in standard English?

Reviewer #1: No

Reviewer #2: Yes

Reviewer #3: Yes

5. Review Comments to the Author

Reviewer #1: The study aimed at elucidating whether hyperacetylation of histones induced by inhibition of HDACs in the testis affect genome wide DNA methylation changes in spermatozoa of mice exposed to VPA

The Manuscript is not suitable for publication in the present form. The comments/suggestions are as follows:

1. The experiments done show upregulation of testicular histone acetylase in the testis and effect on Genome-wide DNA methylation, however, it has failed to demonstrate if the effect on DNA methylation is due to hyperacetylation of histones. Showing by in-silico analysis genes with hypermethylated CpGs in their promoters regulated by HDAC does not suggest that DNA methylation is regulated by HDAC.

2. VPA is known to inhibit both class I and II HDAC family including HDAC 6. HDAC 6 is known to deacetylase tubulin and hence some of the effects of VPA on spermatogenesis or sperm function could be due to effect of tubulin acetylation which is very well reported.

3. It is not clear why only H3K9 and H3K27 acetylation was studied. H4 acetylation is important for chromatin remodeling during spermiogenesis. It would more informative to do pan H3 acetylation and H4 acetylation.

4. It is well evident in literature that histones are acetylated in spermatogonia to preleptotene and then in elongating spermatids. Histones are underacetylated during meiosis and post meiosis (round spermatid). Hence, hyperacetylation of histones due to VPA treatment would affect meiosis and early spermiogenesis. It would have been interesting to study histone acetylation status in different spermatogenic cell types in the testis following VPA treatment.

5. Age of mice used for the study was 6 to 10 weeks, this is too broad a range. Was the duration of treatment same for all them, i.e. 4 weeks?

6. The conclusion that, “These findings suggest that VPA-treated mice may be a useful, simple epigenetic, intergenerational inheritance model for histone hyperacetylation”. The results of the study do not support this conclusion.

Reviewer #2: I read the article with interest. The authors have tested the effect of valporic acid on sperm DNA methylation and concluded that valporic acid results in DNA methylation changes by histone acetylation. Valporic acid is under in the treatment of epilepsy and hence may be tested for transgenerational toxicity, which makes the basis of this study. While the authors have some shown some behavioural changes in the next generation, they were under specific conditions. I have the following comments;

1. Why were behavioural tests important in this study? Since the aim is to detect transgenerational DNA methylation changes, efforts should be made to gain insights in DNA methylation

2. Regarding DNA methylation changes, the authors mention that they would like to test for inheritance of changes upto the F2 generation. But in the results described, they have compared F0 and F1 generations for DNA methylation changes. While all DNA methylation changes in the F0 cannot be expected to inherit to the F1 generation, I could not find data for F2 generation. This paragraph ends with the comparison of F0 and F1. In such a case, it cannot be concluded that the DNA methylation changes are transgenerational.

Reviewer #3: It is a relevant, well written and clear article. It is a very complementary work to various studies published in recent years on epigenetic aspects and more particularly DNA methylation. In this article the scientific question posed in health terms is essential and the methodology is appropriate. Indeed, exposure to chemicals in the population and more particularly during development is a subject of societal and clinical questioning and more particularly for the reproductive system.

This is a very good and original preclinical work that clearly shows that DNA methylation is a key player in epigenetic mechanisms transmitted through generations associated with the impact of environmental exposure.

For this article to be publishable in PLOS ONE, I recommend that the authors work on the following points:

-Introduction: define epigenetics and the mechanisms of action of DNA methylation. Add a summary table of different studies.

-Results: Information on the clinical condition of the animals is missing (general condition, weight, clinical biological analysis)

Discussion: too long, bring out the essential, talk about the impact of exposure on offspring and putative epigenetic mechanisms on reproduction during pre- and post-natal development and their potential health consequences. To propose in a clearer way perspective in terms of future research.

6. PLOS authors have the option to publish the peer review history of their article (what does this mean?). If published, this will include your full peer review and any attached files.

Reviewer #1: No

Reviewer #2: **Yes: **Rajender Singh

Reviewer #3: **Yes: **Maâmar Souidi

---

## [Author Response · Author response to Decision Letter 0]

3 Jan 2023

Dear Editors and Reviewers

Thank you very much for reviewing our manuscript and offering valuable advice.

We have addressed your comments with point-by-point responses, and revised the manuscript accordingly.

Response to reviewer 1:

We wish to express our appreciation to the Reviewer for his or her insightful comments, which have helped us significantly improve the paper.

Comment 1: The experiments done show upregulation of testicular histone acetylase in the testis and effect on Genome-wide DNA methylation, however, it has failed to demonstrate if the effect on DNA methylation is due to hyperacetylation of histones. Showing by in-silico analysis genes with hypermethylated CpGs in their promoters regulated by HDAC does not suggest that DNA methylation is regulated by HDAC.

Response: Thank you very much for your important comments. To demonstrate that testicular histone hyperacetylation by HDAC inhibition is associated with altered sperm DNA methylation, we used public ChIP-seq data provided by ChIP-Atlas (https://chip-atlas.org/) to search for acetylated histone H3(K9 and K27)-rich genomic regions in the vicinity of DMRs. Our results revealed the presence of acetylated histone H3 in the vicinity of many DMRs, suggesting that the methylation changes observed in our experiments are associated with the acetylation of histone H3. Based on this finding, we modified Fig. 2D to show the regions of acetylated histones H3K9 and K27 near the DMRs in the ideogram. Along with this modification, we added the following text to the method, result, figure legend, and discussion sections:

“The genomic regions rich in acetylated histones K9 and K27 located in the vicinity of the DMRs (± 1 kb) were identified using the Peak Browser of ChIP-Atlas [32], where the threshold for significance was set to 50 (q-value < 1E-05) and cell type as mouse male germ line (testis, male germ cells, spermatogonia, spermatogenic cells, round spermatids, and spermatids).” (page 6, lines 174-178)

“We then searched for acetylated histone H3(K9 and K27)-rich genomic regions in the vicinity of these DMRs (±1 kb) using public ChIP-seq data provided by ChIP-Atlas and found that many DMRs were in the vicinity of acetylated histone H3 (Fig 2D).” (page 10, lines 295-297.)

“(D) Positions of differentially methylated regions (DMRs) found in the VPA-treated group compared with those of the control group are indicated by triangle markers on the ideogram. The positions of acetylated histone H3 (K9 and K27) which have been reported to be present in the vicinity of DMRs (±1 kb) in the male germline are also indicated by rectangular markers on the ideogram. The regions that contained at least 10 CpGs within 300 bp and whose average difference in methylation rate was ≥ 20% were extracted as DMRs. Genomic regions with a score of 50 or higher in the ChIP-Atlas Peak Browser were designated as acetylated histone regions.” (page 10, lines 309-315)

“Many of the DMRs we found are located near acetylated histone regions, suggesting that HDAC inhibition by VPA is linked to DNA methylation changes.” (page 13, lines 407-409)

Comment 2: VPA is known to inhibit both class I and II HDAC family including HDAC 6. HDAC 6 is known to deacetylase tubulin and hence some of the effects of VPA on spermatogenesis or sperm function could be due to effect of tubulin acetylation which is very well reported

Response: We thank the reviewer for this comment. As you pointed out, it is possible that an overdose of VPA could affect spermatogenesis and sperm function. However, since we administered VPA at concentrations and for durations that did not cause testicular toxicity, we do not believe that our results were affected by acetylation of sperm tubulin. This can be explained by the fact that in an experiment in which F1 mice were produced by natural mating, the control group and the VPA-treated group had approximately the same number of litters, and all litters were healthy.

Comment 3: It is not clear why only H3K9 and H3K27 acetylation was studied. H4 acetylation is important for chromatin remodeling during spermiogenesis. It would more informative to do pan H3 acetylation and H4 acetylation.

Response: We thank the Reviewer for this pertinent comment. We examined histone H3K9 and K27 as representative histone acetylation sites that are more closely related to gene expression regulation and more widely studied. Since VPA is known to increase overall histone acetylation, including histone H4 (Marchion et al., Cancer Res., 2005. 65(9):3815-22), it is expected that an increase in overall histone acetylation is occurring in this study. As the reviewer pointed out, histone H4 modification is important for chromatin remodeling and may be involved in the phenomenon of epigenomic inheritance via residual histones in sperm, which we intend to make one of our next research topics.

Comment 4: It is well evident in literature that histones are acetylated in spermatogonia to preleptotene and then in elongating spermatids. Histones are underacetylated during meiosis and post meiosis (round spermatid). Hence, hyperacetylation of histones due to VPA treatment would affect meiosis and early spermiogenesis. It would have been interesting to study histone acetylation status in different spermatogenic cell types in the testis following VPA treatment.

Response: Thank you very much for your helpful comments. We previously attempted to compare the intensity of histone acetylation in each differentiation stage of spermatogenic cells using immunostaining, but this method lacked quantitative properties. Therefore, we chose to use western blotting method to compare acetylation in the whole testis. Other methods could be considered to compare the acetylation status of histones after fractionation of cells by FACS, but in this study, we thought that observing the epigenetic status of sperm would be a better first step. The reviewer's suggestion is very important and our next task in investigating how the epigenetic changes in sperm observed in this study occurred.

Comment 5: Age of mice used for the study was 6 to 10 weeks, this is too broad a range. Was the duration of treatment same for all them, i.e. 4 weeks?

Response: We apologize for our insufficient explanation. All mice began receiving VPA or saline at 6 weeks of age and continued for 4 weeks until 10 weeks of age. We had incorrectly described it and have corrected it as follows:

“VPA and saline were administered daily via intraperitoneal injection to mice for 4 weeks from 6 to 10 weeks of age.” (page 3, lines 88-89.)

Comment 6: The conclusion that, “These findings suggest that VPA-treated mice may be a useful, simple epigenetic, intergenerational inheritance model for histone hyperacetylation”. The results of the study do not support this conclusion.

Response: Thank you for your comment. As the reviewer pointed out, this is a matter to be written in DISCUSSION, not in conclusion. Therefore, the following modifications have been made:

“These findings suggest that VPA-induced histone hyperacetylation may have brain function-related effects on the next generation through changes in sperm DNA methylation.” (page 1, lines 25-26.)

Response to reviewer 2:

We wish to express our strong appreciation to the reviewer for his or her insightful comments on our paper.

Comment 1: Why were behavioural tests important in this study? Since the aim is to detect transgenerational DNA methylation changes, efforts should be made to gain insights in DNA methylation.

Response: We appreciate the reviewer's comment on this point. Many studies of epigenetic inheritance have reported cases in which some stimulus received in the parental generation affects the traits of the next generation, including brain function. Thus, we employed behavioral experiments to comprehensively examine phenotypically expressed traits. As the reviewer pointed out, more effort should be directed to DNA methylation to examine "transgenerational" effects, which is our next challenge. However, we chose this experimental design because we wanted to examine "intergenerational" effects via the sperm epigenome.

Comment 2: Regarding DNA methylation changes, the authors mention that they would like to test for inheritance of changes upto the F2 generation. But in the results described, they have compared F0 and F1 generations for DNA methylation changes. While all DNA methylation changes in the F0 cannot be expected to inherit to the F1 generation, I could not find data for F2 generation. This paragraph ends with the comparison of F0 and F1. In such a case, it cannot be concluded that the DNA methylation changes are transgenerational.

Response: Thank you very much for your helpful comments. Epigenetic transgenerational inheritance is defined as “the germline (egg or sperm) transmission of epigenetic information between generations in the absence of any environmental exposure” (Skinner KM, BMC Med, 2014, 12(153)). In our experimental design, the F1 generation could be considered to have already been exposed to VPA as an effect on the sperm of the F0 generation. Therefore, to examine the transgenerational effects of VPA, it is necessary to examine the F2 generation, which is not exposed to VPA. However, we could not generate the F2 generation because we needed to examine the brains of F1 mice immediately after the behavioral experiments to investigate the intergenerational effects of VPA. Therefore, we examined sperm DNA methylation in F1 mice to investigate the possibility of transmission to the F2 generation.

Response to reviewer 3:

Thank you for your valuable suggestions on our manuscript.

Comment 1: Introduction: define epigenetics and the mechanisms of action of DNA methylation. Add a summary table of different studies.

Response: Thank you very much for your helpful comments. We apologize for missing the basic definitions of epigenetics and DNA methylation. The following text has now been added:

“Epigenetics is the study of mitotically or meiotically heritable changes in gene expression or cellular phenotypes that occur without altering DNA sequences [15].” (page 2, lines 52-53.)

“DNA methylation is the most well-studied epigenetic modification in mammals and primarily involves the addition of a methyl group to the C5 position of cytosine. DNA methylation plays a role in regulating gene expression by recruiting proteins involved in repressing gene expression and by inhibiting the binding of transcription factors to DNA [21].” (pages 2-3, lines 62-65.)

In addition, our current study is not focused on investigating VPA as one of the environmental chemicals that show toxic effects on the next generation, but rather on investigating the effects of VPA as a model compound to elucidate the mechanism of such epigenetic intergenerational phenomena. In fact, the discussion section proposes an experimental design in which the environmental stimuli are combined with the VPA in order to examine the epigenetic inheritance of the environmental stimuli. Therefore, since the chemicals and their effects reported in other studies should not be discussed in parallel with VPA, and presenting them in a table may confuse the reader, we made the decision not to create a summary table in this study.

Comment 2: Results: Information on the clinical condition of the animals is missing (general condition, weight, clinical biological analysis)

Response: We apologize for our insufficient explanation about the animals. All animals were healthy during the period of VPA or saline administration. Unfortunately, we did not weigh the animals at autopsy, but no differences in testis weight were observed. The F1 generation mice used in the behavioral experiments have already been described and were healthy throughout the experimental period, with no differences in body weight between groups. For the clinical condition of the mice, we have added the following text to the result section:

“During the period of VPA or saline administration, all animals were healthy and had no clinical abnormalities, including testicular toxicity.” (page 9, lines 264-265.)

Comment 3: Discussion: too long, bring out the essential, talk about the impact of exposure on offspring and putative epigenetic mechanisms on reproduction during pre- and post-natal development and their potential health consequences. To propose in a clearer way perspective in terms of future research.

Response: We appreciate the reviewer’s suggestions. We have now omitted the following sentences from the DISCUSSION to make it more concise:

“These DMCs may contain false positives because we used P-values rather than q-values to determine the significance in this experiment. However, chemical-induced epigenetic changes often induce more subtle changes rather than large methylation changes, and such subtle changes lead to changes in behavioral traits [23]. In addition, owing to the large number of CpGs used in the calculations (over 37 million), we considered the risk of missing false negatives due to FDR calculations and determined DMC with p < 0.05 and an average methylation rate change of at least 20%.”

“However, this behavioral change was modest, given that no difference was observed in the center time in the open field test.”

“We analyzed the effects of VPA administration at the molecular level on the cell lineage.”

“The system is useful in elucidating epigenetic intergenerational phenomena since it uses only chemical administration.”

In addition, we have added the following sentences to simply state our hypothesis for the phenomena observed in this study and our outlook for future research:

“We believe that this phenomenon is due to the synergistic effects of chromatin relaxation, which promotes epigenetic changes, and the pharmacological effects of VPA itself, such as HDAC inhibition and involvement in GABA signaling.” (pages 14-15, lines 446-448.)

We wish to thank the Reviewer again for his or her valuable comments.

---

## [Decision Letter · Decision Letter 1]

19 Jan 2023

PONE-D-22-26282R1Testicular histone hyperacetylation in mice by valproic acid administration affects the next generation by changes in sperm DNA methylationPLOS ONE

Dear Dr. Tanemura,

Thank you for submitting your manuscript to PLOS ONE. After careful consideration, we feel that it has merit but does not fully meet PLOS ONE’s publication criteria as it currently stands. Therefore, we invite you to submit a revised version of the manuscript that addresses the points raised during the review process. Concerns still exist on the use of morulas to analyze epigenetic methylation and also the relevance between F1 and F2 generation spermatozoa, in light of epigenetic reprogramming that occurs. You are advised to provide additional data / justification on the logic of using samples at the particular stages.

We look forward to receiving your revised manuscript.

Kind regards,

Suresh Yenugu

Academic Editor

PLOS ONE

Reviewers' comments:

Reviewer's Responses to Questions

**Comments to the Author**

1. If the authors have adequately addressed your comments raised in a previous round of review and you feel that this manuscript is now acceptable for publication, you may indicate that here to bypass the “Comments to the Author” section, enter your conflict of interest statement in the “Confidential to Editor” section, and submit your "Accept" recommendation.

Reviewer #1: (No Response)

Reviewer #2: (No Response)

Reviewer #3: All comments have been addressed

2. Is the manuscript technically sound, and do the data support the conclusions?

Reviewer #1: Partly

Reviewer #2: Yes

Reviewer #3: Yes

3. Has the statistical analysis been performed appropriately and rigorously? 

Reviewer #1: Yes

Reviewer #2: Yes

Reviewer #3: Yes

4. Have the authors made all data underlying the findings in their manuscript fully available?

Reviewer #1: Yes

Reviewer #2: Yes

Reviewer #3: Yes

5. Is the manuscript presented in an intelligible fashion and written in standard English?

Reviewer #1: Yes

Reviewer #2: Yes

Reviewer #3: Yes

6. Review Comments to the Author

Reviewer #1: All comments raised has been satisfactorily addressed except the first on which the conclusion is based. The authors conclude that, "These findings suggest that VPA-induced histone hyperacetylation may have brain function-related effects on the next generation through changes in sperm DNA methylation. Demonstrating presence of acetylated histone H3 in the vicinity of the DMR based on available dataset does not suggest that histone hyperacetylation affects DNA methylation. In fact, recent reports suggests that "VPA as an HDAC inhibitor does not induce changes only in histone acetylation, but also changes in the state of DNA modification. It shows cross-reactivity between chromatin remodeling due to histone acetylation and DNA methylation" ( Cross-reactivity between histone demethylase inhibitor valproic acid and DNA methylation in glioblastoma cell lines Anna-Maria Barciszewska et al; Front Oncol. 2022 Nov 16;12:1033035. doi: 10.3389/fonc.2022.1033035.) The authors need to include this point in the manuscript

Reviewer #2: I have the following questions, to which I would like to have answers.

1. Sperm DNA extracted for methylation analysis from VPA-treated mice was of which day after treatment- day 0, 3, 7?

2. They have used morula stage for checking methylation level for reference to F1 generation.

Why morula stage?

DNA undergoes two time epigenetic reprogramming, first immediate fertilization and second at embryo7.25 stage in PGCs. So morula has just crossed first epigenetic reprogramming.

3. Similarly, it goes for F1 generation sperm, that these sperm cannot be used to depict methylation level of F2 generation as it has to undergo embryonic epigenetic reprogramming.

4. In behavioural test (fear), conditioning for 1 day does not seem to be sufficient also why behavioural test has been done only in F1 generation and not in F0.

Reviewer #3: reproductive system.

This is a very good and original preclinical work that clearly shows that DNA methylation is a key player in epigenetic mechanisms transmitted through generations associated with the impact of environmental exposure.

7. PLOS authors have the option to publish the peer review history of their article (what does this mean?). If published, this will include your full peer review and any attached files.

Reviewer #1: No

Reviewer #2: **Yes: **Rajender Singh

Reviewer #3: **Yes: **Maâmar Souidi

---

## [Author Response · Author response to Decision Letter 1]

6 Feb 2023

Dear Editors and Reviewers

Thank you very much for reviewing our manuscript and offering valuable advice.

We have addressed your comments with point-by-point responses, and revised the manuscript accordingly.

Response to reviewer 1:

 We wish to express our appreciation to the Reviewer for his or her insightful comments, which have helped us significantly improve the paper.

Comment: All comments raised has been satisfactorily addressed except the first on which the conclusion is based. The authors conclude that, "These findings suggest that VPA-induced histone hyperacetylation may have brain function-related effects on the next generation through changes in sperm DNA methylation. Demonstrating presence of acetylated histone H3 in the vicinity of the DMR based on available dataset does not suggest that histone hyperacetylation affects DNA methylation. In fact, recent reports suggests that "VPA as an HDAC inhibitor does not induce changes only in histone acetylation, but also changes in the state of DNA modification. It shows cross-reactivity between chromatin remodeling due to histone acetylation and DNA methylation" ( Cross-reactivity between histone demethylase inhibitor valproic acid and DNA methylation in glioblastoma cell lines Anna-Maria Barciszewska et al; Front Oncol. 2022 Nov 16;12:1033035. doi: 10.3389/fonc.2022.1033035.) The authors need to include this point in the manuscript.

Response: We appreciate the helpful suggestion of the Reviewer. Although the report presented by the reviewer is the result of glioblastoma cells only, we also believe that it is a very important point. Within the manuscript, we also mention that the effect of VPA itself on DNA methylation should be considered (lines 446-448:” We believe that this phenomenon is due to the synergistic effects of chromatin relaxation, which promotes epigenetic changes, and the pharmacological effects of VPA itself, such as HDAC inhibition and involvement in GABA signaling.”). However, thanks to the reviewer's comments, we recognized the need for more detailed mention and have added the following text to the Discussion section:

“Since a recent study suggests that VPA affects DNA modification independently of histone acetylation, future studies should validate our results with HDAC inhibitors other than VPA, such as trichostatin A [44].” (page 14, lines 450-51.)

Response to reviewer 2:

 We wish to express our strong appreciation to the reviewer for his or her insightful comments on our paper.

Comment 1: Sperm DNA extracted for methylation analysis from VPA-treated mice was of which day after treatment- day 0, 3, 7?

Response: We performed sperm DNA methylation analysis using mice two weeks after the last dose date. We apologize for our insufficient explanation and have added the following sentence to the relevant section:

“Two weeks after the final dose, 38 animals were sacrificed at 12 weeks of age and used for in vitro fertilization and sperm DNA methylation analysis.” (page 3, lines 90-92.)

Comment 2: They have used morula stage for checking methylation level for reference to F1 generation. Why morula stage? DNA undergoes two time epigenetic reprogramming, first immediate fertilization and second at embryo7.25 stage in PGCs. So morula has just crossed first epigenetic reprogramming.

Response: The reviewer correctly pointed out the important matter. If sperm DNA methylation is transmitted to the next generation, it should remain against epigenetic reprogramming immediately after fertilization. Therefore, we wanted to examine fertilized eggs immediately after fertilization, but we performed IVC because the number of cells (= amount of DNA) for analysis is very small. Meanwhile, as fertilized eggs are grown to blastocysts and beyond, the next generation's unique DNA methylation is established, making the interpretation of results at this stage more difficult. Therefore, we performed our analysis at the morula stage, a stage in which the number of cells is high and DNA methylation of parental origin appears to have been eliminated.

Comment 3: Similarly, it goes for F1 generation sperm, that these sperm cannot be used to depict methylation level of F2 generation as it has to undergo embryonic epigenetic reprogramming.

Response: As stated in the manuscript, it has been suggested that some regions escape epigenetic reprogramming and be transmitted to the next generation. The phenomenon of conservation of DNA methylation changes at some loci over multiple generations has been reported (Belleau P et al., Inferring and modeling inheritance of differentially methylated changes across multiple generations, Nucleic Acids Research. 2018. https://doi.org/10.1093/nar/gky362). It is also well known that epigenomic changes in sperm can cause phenotypic changes over multiple generations, and our results are important for investigating the mechanisms of such phenomena.

Comment 4: In behavioural test (fear), conditioning for 1 day does not seem to be sufficient also why behavioural test has been done only in F1 generation and not in F0.

Response: Our fear conditioning follows the general protocol and one day of conditioning is sufficient (Curzon P, Rustay NR, Browman KE. Cued and Contextual Fear Conditioning for Rodents. In: Buccafusco JJ, editor. Methods of Behavior Analysis in Neuroscience. 2nd edition. Boca Raton (FL): CRC Press/Taylor & Francis; 2009. Chapter 2. Available from: https://www.ncbi.nlm.nih.gov/books/NBK5223/).

VPA has been reported to affect memory performance through its pharmacological effects (Sintoni S et al., Chronic valproic acid administration impairs contextual memory and dysregulates hippocampal GSK-3β in rats. Pharmacology Biochemistry and Behavior. 2013 https://doi.org/10.1016/j.pbb.2013.02.013.). Therefore, we did not perform behavioral experiments on F0 mice, as this would only make interpretation of the experimental results more difficult.

Again, thank you for giving us the opportunity to strengthen our manuscript with your valuable comments and queries. We have worked hard to incorporate your feedback and hope that these revisions persuade you to accept our submission.

---

## [Decision Letter · Decision Letter 2]

27 Feb 2023

Testicular histone hyperacetylation in mice by valproic acid administration affects the next generation by changes in sperm DNA methylation

PONE-D-22-26282R2

Dear Dr. Tanemura,

We’re pleased to inform you that your manuscript has been judged scientifically suitable for publication and will be formally accepted for publication once it meets all outstanding technical requirements.

Kind regards,

Suresh Yenugu

Academic Editor

PLOS ONE

Additional Editor Comments (optional):

Reviewers' comments:

Reviewer's Responses to Questions

**Comments to the Author**

1. If the authors have adequately addressed your comments raised in a previous round of review and you feel that this manuscript is now acceptable for publication, you may indicate that here to bypass the “Comments to the Author” section, enter your conflict of interest statement in the “Confidential to Editor” section, and submit your "Accept" recommendation.

Reviewer #1: All comments have been addressed

Reviewer #2: All comments have been addressed

2. Is the manuscript technically sound, and do the data support the conclusions?

Reviewer #1: Partly

Reviewer #2: Yes

3. Has the statistical analysis been performed appropriately and rigorously? 

Reviewer #1: Yes

Reviewer #2: Yes

4. Have the authors made all data underlying the findings in their manuscript fully available?

Reviewer #1: Yes

Reviewer #2: Yes

5. Is the manuscript presented in an intelligible fashion and written in standard English?

Reviewer #1: Yes

Reviewer #2: Yes

6. Review Comments to the Author

Reviewer #1: The Comments raised have been satisfactorily responded by the authors

The MS is now suitable for publication

Reviewer #2: The authors have addressed all my comments and the manuscript may be accepted now. I do not have any further comments.

7. PLOS authors have the option to publish the peer review history of their article (what does this mean?). If published, this will include your full peer review and any attached files.

Reviewer #1: No

Reviewer #2: **Yes: **Rajender Singh

---

## [Editor Report · Acceptance letter]

1 Mar 2023

PONE-D-22-26282R2 

Testicular histone hyperacetylation in mice by valproic acid administration affects the next generation by changes in sperm DNA methylation 

Dear Dr. Tanemura:

I'm pleased to inform you that your manuscript has been deemed suitable for publication in PLOS ONE. Congratulations! Your manuscript is now with our production department. 

Kind regards, 

on behalf of

Dr. Suresh Yenugu 

Academic Editor

PLOS ONE